# Pitting Corrosion and Microstructure of J55 Carbon Steel Exposed to CO_2_/Crude Oil/Brine Solution under 2–15 MPa at 30–80 °C

**DOI:** 10.3390/ma11122374

**Published:** 2018-11-26

**Authors:** Haitao Bai, Yongqing Wang, Yun Ma, Peng Ren, Ningsheng Zhang

**Affiliations:** 1Institute of Petroleum and Gas Engineering, Southwest Petroleum University, Chengdu 610500, China; baihaitao_xsyu@126.com; 2College of Petroleum Engineering, Key Laboratory of Environment Pollution Control Technology of Oil Gas and Reservoir Protection in Shaanxi Protection, Xi’an Shiyou University, Xi’an 710065, China; mayun_xian123@126.com (Y.M.); zhangnsh_xian@126.com (N.Z.); 3Xi’an Taijin Industrial Electrochemical Technology Co., Ltd., Xi’an 710016, China; renpeng_xian@126.com

**Keywords:** CO_2_ corrosion, J55 carbon steel, CO_2_/crude oil/brine mixtures, corrosion depth distribution

## Abstract

This study aimed to evaluate the corrosion properties of J55 carbon steel immersed in CO_2_/crude oil/brine mixtures present in the wellbores of CO_2_-flooded production wells. The main corroded position of wellbore was determined and wellbore corrosion law was provided. Corrosion tests were performed in 30% crude oil/brine solution under the simulated temperature (30–80 °C) and pressure (2–15 MPa) conditions of different well depths (0–1500 m). The corrosion behavior of J55 carbon steel was evaluated through weight-loss measurements and surface analytical techniques, including scanning electron microscopy, energy dispersive spectrometer, X-ray diffraction analysis, and optical digital microscopy. Corrosion rate initially increased and then decreased with increasing well depth, which reached the maximum value of 1050 m. At this well depth, pressure and temperature reached 11 MPa and 65 °C, respectively. Under these conditions, FeCO_3_ and CaCO_3_ localized on sample surfaces. Microscopy was performed to investigate corrosion depth distribution on the surfaces of the samples.

## 1. Introduction

CO_2_ is internationally recognized as a major greenhouse gas that accounts for approximately 65% of the total greenhouse gas emissions [1,2,3]. All countries currently attach considerable importance to environmental issues, particularly global warming caused by CO_2_. CO_2_ is used as an oil-flooding agent worldwide because it can effectively reduce crude oil viscosity and residual oil saturation, dissolve gum in reservoirs, and increase permeability and crude oil recovery rate [4,5]. CO_2_ flooding can reduce CO_2_-associated air pollution and greenhouse effects. Nevertheless, oil pipe failure caused by CO_2_ corrosion has become a commonly encountered problem in oilfields and results in great economic losses and safety hazards while severely restricting the development of CO_2_ flooding technology [4,5,6].

CO_2_ corrosion and its control in oil casing and surface transmission pipelines have been important topics in the field of oil and gas exploration. Related studies have focused on the influence of environmental and material factors on corrosion behavior [7,8,9,10,11,12,13,14,15,16]. For example, the corrosive medium and environment have been identified as the deciding factors of corrosion rate and morphology. The influence of temperature and pressure on corrosion rate is mainly reflected by the changes that they induce in the protectiveness of the corrosion product layer [7,8,9,10,11,12,13,14,15,16,17,18]. These changes, in turn, lead to changes in the corrosion rate. With increasing temperature, the CO_2_ corrosion rate of carbon steel initially increases and then decreases [8,9], and the maximum value of CO_2_ corrosion rate changes. Li [10] stated that the maximum corrosion rate of P110 tubular steel is attained at 100 °C. The CO_2_ corrosion rate of carbon steel increases under increasing CO_2_ pressure [11,12,13]. Choi [14], however, proposed that the corrosion rates of carbon steel (API 5CT L80) in 25 wt.% NaCl solution at 65 °C negligibly changes with pressure (4, 8, and 12 MPa). The CO_2_ corrosion rate of carbon steel increases with the water cut of water/crude oil mixtures [15,16,17,18]. Nevertheless, efficient anticorrosion measures cannot be established on the basis of the results of preliminary studies that have largely focused on the influence of a single factor on CO_2_ corrosion and of the limited works that have investigated the laws governing CO_2_ corrosion in the wellbores of CO_2_-flooded wells. Although studies on aqueous CO_2_ corrosion in crude oil/brine mixtures have been performed, comprehensive studies on CO_2_ corrosion in CO_2_/crude oil/brine mixtures at different well depths remain unavailable.

This work explored the laws governing CO_2_ corrosion at different depths of CO_2_-flooded wells. Specifically, this study investigated the common effects of temperature and pressure on CO_2_ corrosion in a CO_2_/crude oil/brine environment. The average corrosion rate was quantified through the weight-loss method. The surface morphologies of corrosion samples were observed through scanning electron microscopy (SEM). The compositions of corroded surfaces were analyzed through energy dispersive Spectroscopy (EDS) and X-ray diffraction (XRD) analyses. Corrosion depth distribution was analyzed through microscopy, 42 percent of the exposed surface area was studied, which can accurately reflect the surface conditions of the samples after corrosion.

## 2. Materials and Methods

### 2.1. Experimental Materials

J55 carbon steel was processed into rectangular samples (50 mm × 10 mm × 3 mm, *φ* = 6 mm) for the weight-loss test and surface analysis. The chemical composition of J55 carbon steel is shown in Table 1. The samples were placed in acetone to remove surface oil and then immersed in ethanol for 5 min for degreasing and dehydration. The samples were collected, dried with cold air, packed in filter paper, and placed in a dryer for 4–7 h. The sizes and weights of the samples were measured within an accuracy of 0.1 mg.

The corrosive medium comprised crude oil and brine. Crude oil was collected from the Chang-8 oil reservoir, a certain block in Changqing Oilfield, and the composition of simulated brine was based on the composition of the water produced in the Chang-8 oil reservoir. The compositions of crude oil and brine are shown in Table 2 and Table 3, respectively.

### 2.2. Weight-Loss Corrosion Test

The corrosion test was performed in accordance with the weight-loss method with a PARR-4578 autoclave. The schematic of the test is shown in Figure 1. 1 L the mixture of crude oil and brine (*v*:*v* = 3:7) was added to the autoclave and purged with a small amount of N_2_ for 120 min under 0.5 MPa to remove dissolved O_2_. Then, the mixture was subjected to 60 min of injection with high-purity CO_2_ under 1 MPa to remove N_2_ [19]. Finally, autoclave temperature was increased to the test temperature, and autoclave pressure was increased to the test pressure with high-purity CO_2_. The test conditions were maintained for 2 days at a running speed of 0.5 m·s^−1^ (200 r·min^−1^).

The corrosion rate of the corroded steel was determined through the mass-loss method in accordance with the *ASTM G1-03 Standard Practice for Preparing, Cleaning, and Evaluating Corrosion Test Specimens* [20]. The samples were immediately rinsed with distilled water. Acetone was used to remove crude oil from the surfaces of the samples after corrosion induction. Then, the samples were immersed in acid cleaning solution (500 mL of HCl and 3.5 g of hexamethylenamine brought to volume with water to 1000 mL) for 10 min. At the same time, the corrosion products were removed from the surfaces of the samples, and the samples were collected from the acid cleaning solution. The acid cleaning solution was thoroughly rinsed off from the surfaces of the samples with distilled water. The samples were then twice immersed in ethanol for cleaning and dehydration, collected, placed on filter paper, dried with cold air, packed in filter paper, and placed in the dryer for 4–7 h. Finally, the samples were weighed to within an accuracy of 0.1 mg. Corrosion rate was calculated using the following formula:(1)$$r_{corr}=\frac{8.76\times 10^{4}\times (m-m_{t})}{S\times t\times \rho }\;$$
where *r_corr_* is average corrosion rate (mm·y^−1^); *m* and *m_t_* are the weights of the test sheet before and after the experiment, respectively (g); *S* is the area of the whole surface in contact with the solution (cm^2^); *ρ* is the density of the tested steel (g·cm^−3^); and *t* is the duration of immersion (h). Each test was performed with three parallel samples. The mean corrosion rate error was calculated on the basis of the results for the three parallel samples.

### 2.3. Microstructure Observation

After corrosion induction, samples were extracted from the autoclave and rinsed with distilled water and acetone. The surface microstructures of the corrosion product layers on the surfaces of corroded samples were analyzed through SEM with FEI Quantu 600F microscope (Hillsboro, OR, USA). The elemental compositions of the corrosion product layers were determined with OXFORD INCA energy 350. The compositions of the corrosion product characterized were determined through XRD by using Bruker D8 XRD (Billerica, MA, USA).

### 2.4. Statistics of Corrosion Depth Distribution

The surface depths of the corroded samples cleaned using acid cleaning solution were visualized using an OLYMPUS DSX500 optical digital microscope (Tokyo, Japan). Sample surfaces were subjected to grand horizon 3D image capture under bright-field mode with the adjacent visual threshold splicing mode. As shown in Figure 2, images for the analysis of surface corrosion morphology were acquired at eight observation points on both sides of the same sample through nine-field splicing under 200× magnification and 10% coincidence rate. The display heights of the 3D images were adjusted to the maximum pitting height to ensure that the different ranges of corrosion depths in the same set of images can be represented by different colors. Identical corrosion depth ranges were represented by the same color. Finally, the 3D images were converted into the contour diagrams of corrosion depth distribution.

## 3. Results and Discussion

### 3.1. Corrosion Law of the Deepening Well

The CO_2_-flooded well in Chang-8 Oil Reservoir of a certain block in Changqing Oilfield was taken as an example. The experimental well had a depth of 1550 m, a well-head temperature of 30 °C, pressure of 2 MPa, and well-bottom temperature of 82 °C. Temperature and pressure distributions as a function of well depth are shown in Figure 3. Tests were performed at 0 m (2 MPa, 30 °C), 240 m (4 MPa, 40 °C), 580 m (7 MPa, 50 °C), 800 m (9 MPa, 55 °C), 1050 m (11 MPa, 65 °C), and 1500 m (15 MPa, 80 °C). The liquid produced by the well had a water cut of 70%. The crude-oil composition of the liquid is shown in Table 2. The chemical composition of water produced by the well is shown in Table 3.

Figure 4 shows the appearance of samples after the removal of corrosion scales. Figure 5 shows the average corrosion rate and the maximum corrosion depth of the samples corroded at different well depths. The average corrosion rate initially increased and then decreased with well depth, except for 1050 m. The general equations for the anodic and cathodic reactions of CO_2_ corrosion in deoxygenated solution are shown as Equations (2) and (3), respectively [9].
(2)$$\;\mathrm{Fe}\to {\mathrm{Fe}}^{2+}+\;2e^{-}\;$$
(3)$$\;{\mathrm{CO}}_{2}{\;+\;H}_{2}O\to {\;H}_{2}{\mathrm{CO}}_{3}\;$$

Temperature and pressure increased as well depth increased. Increasing temperatures reduced the viscosity and protective effect of crude oil on the sample surfaces while intensifying mass transfer between the samples and corrosive medium and accelerating corrosion. Moreover, CaCO_3_ and FeCO_3_ deposits generated from the reaction between CO_3_^2−^ and HCO_3_^−^ and between Ca^2+^ and Fe^2+^ in the liquid gradually increased and inhibited corrosion development by forming a protective layer on the sample surfaces [8,9]. The increase in CO_2_ pressure reduced system pH and is conducive for the formation of the protective corrosion product layer [21]. Hence, corrosion acceleration and inhibitory effects simultaneously occurred in the system. At 580 m, the maximum corrosion depth sharply increased, and corrosion type shifted from uniform to local corrosion because the corrosive environment transformed from a CO_2_/crude oil/brine environment to a subcritical CO_2_/crude oil/brine environment [22]. The scaling ability of the solution decreased, the surface of the sample could not be completely covered by precipitates, and the progress of the anodic reaction could only be partially prevented [23]. These effects resulted in local corrosion. Therefore, the average corrosion rate was low (1.7658 mm·year^−1^), whereas the maximum corrosion depth was high (164.358 μm). Under increasing temperature and pressure, the corrosive environment transformed to the supercritical CO_2_/crude oil/brine environment and was dominated by corrosion. Hence, the maximum average corrosion rate was observed at the depth of 1050 m. The possible reduction in the base level of corrosion depth measurement may have reduced the maximum corrosion depths of the samples corroded at 1050 m. With the further increase in well depth, the pH of the corrosion system became almost constant, and the buildup of CaCO_3_ and FeCO_3_ deposits on the sample surfaces could suppress anodic dissolution [23]. The average corrosion rate decreased. Under dynamic conditions, however, the uneven coverage of the sample surfaces with CaCO_3_ and FeCO_3_ deposits resulted in local corrosion and increased the maximum corrosion depth of the samples corroded at 1500 m.

### 3.2. Microstructures and Compositions of Corrosion Scales

The SEM images of the samples after corrosion at different temperatures and pressures are shown in Figure 6. The results for the spectral analysis of the corroded sample surfaces are shown in Table 4. Before CO_2_ reached a supercritical state, the samples exhibited dense surface coatings that mainly consisted of FeC_3_ and FeCO_3_ and almost lacked CaCO_3_ [14,17,19,24]. The average corrosion rates and maximum corrosion depths of the samples were low. After CO_2_ reached the supercritical state, loose surface coatings that mainly consisted of FeCO_3_ and CaCO_3_ formed and failed to provide effective surface protection to the samples [14,19,24]. Thus, the average corrosion rates and maximum corrosion depths of the samples increased.

Figure 7 shows the XRD spectra of the surface layers of the corroded samples immersed in CO_2_/crude oil/brine mixtures under the given temperatures and pressures at different well depths. FeCO_3_ is the main product of the CO_2_ corrosion of carbon steel [7,8,9,10,11,12,13,14,15,16,17,18,19,25]. Similarly, the corrosion product layer that formed on samples immersed in CO_2_/crude oil/brine mixtures mainly comprised CaCO_3_ and FeCO_3_ complex salts. The specific composition of the product layer may be attributed to the isomorphous substitution of metal cations during CO_2_ corrosion [23]. When [Fe^2+^] × [CO_3_^2−^] in the medium exceeded the FeCO_3_ solubility product *K_sp_* (FeCO_3_), that is, when the FeCO_3_ supersaturation in the medium was S = {[Fe^2+^] × [CO_3_^2−^]}/{*K_sp_* (FeCO_3_)} > 1, FeCO_3_ was deposited on the surfaces of the samples. FeCO_3_ deposition can be represented as follows [26]:Fe^2+^ + CO_3_^2−^ → FeCO_3_ (s)(4)

Ca^2+^ in the solution replaced Fe^2+^ in the FeCO_3_ crystal and ultimately formed the Fe(Ca)CO_3_ complex:Ca^2+^ + FeCO_3_ (s) → Fe^2+^ + CaCO_3_ (s)(5)

### 3.3. Law of Corrosion Depth Distribution

Figure 8 shows the results for the corrosion depth analysis of the sample corroded in 30% crude oil/brine at the well depth of 800 m. Figure 8a,b were obtained through optical digital microscopy under 200× magnification with nine-field splicing and 10% coincidence rate. Figure 8c shows the contour diagram of the corrosion depth distribution of Figure 8a, which was transformed from Figure 8b. The size of one contour diagram of corrosion depth distribution was 7612 μm × 7612 μm, and the total image area observed was 57.94 mm^2^, which accounted for 42% of the sample surface area. Therefore, this corrosion depth analysis method can accurately reflect the surface conditions of the samples after corrosion. Figure 9 shows the contour diagrams of corrosion depth distribution at different observation positions after the removal of corrosion scales from the sample corroded in 30% crude oil/brine at the well depth of 800 m.

The frequency density distribution of corrosion depth on the surface of the sample corroded at the well depth of 800 m is shown in Figure 10. The class intervals used to represent corrosion depth ranges in Figure 10 are the same as colors used to represent corrosion depth ranges in Figure 9. The plot of frequency density was bell-shaped with bilateral symmetry, wherein high values clustered in the center of the plot and low values clustered at both ends of the plot; these characteristics are indicative of typical Gaussian distribution [27,28]. The frequency density distribution maps of samples corroded at different well depths shown in Figure 11 exhibit similar characteristics of Figure 10.

Table 5 shows the fitting parameters for the curve shown in Figure 11. The parameters were obtained by using the Gaussian model (Equation (6)) multicurve global mode in origin 9.0 and a correlation coefficient of 0.9886. Corrosion depths followed Gaussian distribution. The physical interpretation of corrosion depth distribution revealed that *y*_0_ = 0 and *A* = 1 in Equation (6).
(6)$$\;y=y_{0}+\frac{A}{w\sqrt{\pi /2}}e^{-2\frac{(x-x_{c}{)}^{2}}{w^{2}}}\;$$
where *y* is the probability density function; *y*_0_ is the offset, *y*_0_ = 0; *A* is the area, where *A* = 1; *x* is corrosion depth (μm); *x_c_* is the expected value (μm); *w* is twice the standard deviation; *x_c_* determines the location of the distribution curve; and *w* determines the amplitude of the distribution curve.

Table 6 shows the results of variance analysis through the Gaussian model used to fit the curve shown in Figure 11. The Prob > F value of less than 0.01 indicates that the frequency density distribution curve in Figure 11 shows Gaussian distribution. In Gaussian distribution, *x_c_* indicates the average value of the random variables and represents the average corrosion depth. The corrosion of the sample surface may have reduced the datum plane during image acquisition. Moreover, the trend followed by *x_c_* with the change in well depth differed from that followed by the average corrosion rate, especially when the average corrosion rate was high. By adding/subtracting a constant value, the frequency density curve for new random variables generated from random variables with Gaussian distribution was transformed into the translated frequency density curve for former variables in the *x*-direction without changing the shape of the frequency density curve. Therefore, the absence or presence of the datum plane during image acquisition will not affect the *w* of the fitting results for Gaussian distribution. The trend followed by *w* with the change in well depth was the same as that followed by the maximum corrosion depth. This similarity indicates the existence of a strong linear correlation between *w* and corrosion type, as illustrated in Figure 12. Moreover, *w* was positively related to the span of the frequency density curve. A small *w* value represents a density distribution curve with a narrow range and limited corrosion depth distribution. These characteristics indicate uniform corrosion. A high *w* value represents a density distribution curve with a broad range and corrosion depth with broad distribution. These characteristics indicate local corrosion. 

## 4. Conclusions

Based on the observed corrosion behavior of J55 carbon steel in CO_2_/30% crude oil/brine mixtures under the simulated conditions of different well depths (0–1500 m), we conclude the following:(1)The average corrosion rate of J55 carbon steel initially increased and then decreased in the CO_2_/crude oil/brine environment as partial CO_2_ pressure increased. Corrosion type shifted from uniform to local corrosion;(2)The main corrosion products on the surfaces of J55 carbon steel were FeCO_3_ and CaCO_3_;(3)The distribution of corrosion depth obeyed Gaussian distribution, and *w* was positively correlated with the maximum corrosion depth.

## Figures and Tables

**Figure 1 materials-11-02374-f001:**
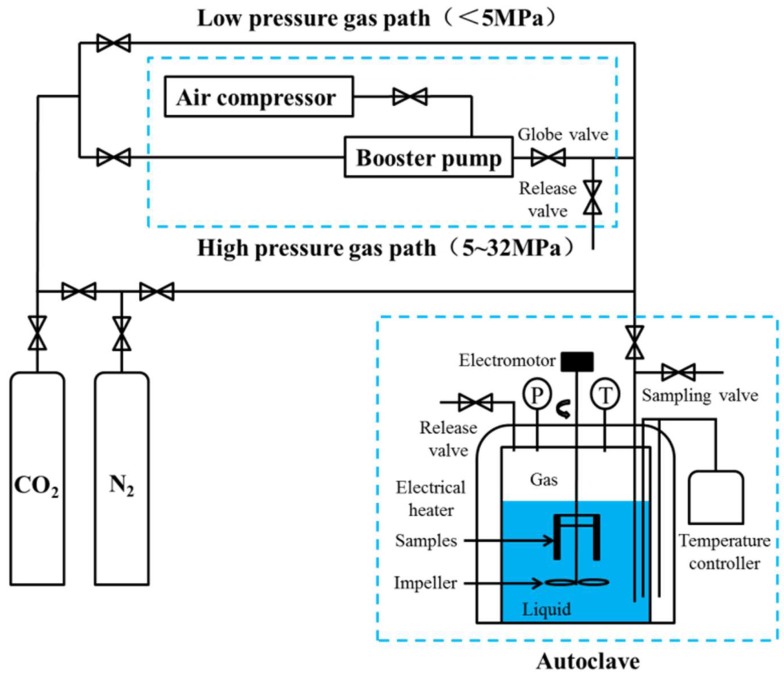
Flow chart of the evaluation system for steel corrosion rate (mass-loss method).

**Figure 2 materials-11-02374-f002:**
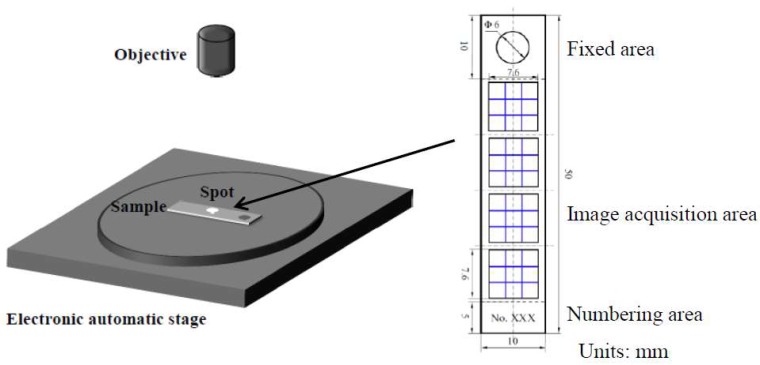
Schematic of image acquisition.

**Figure 3 materials-11-02374-f003:**
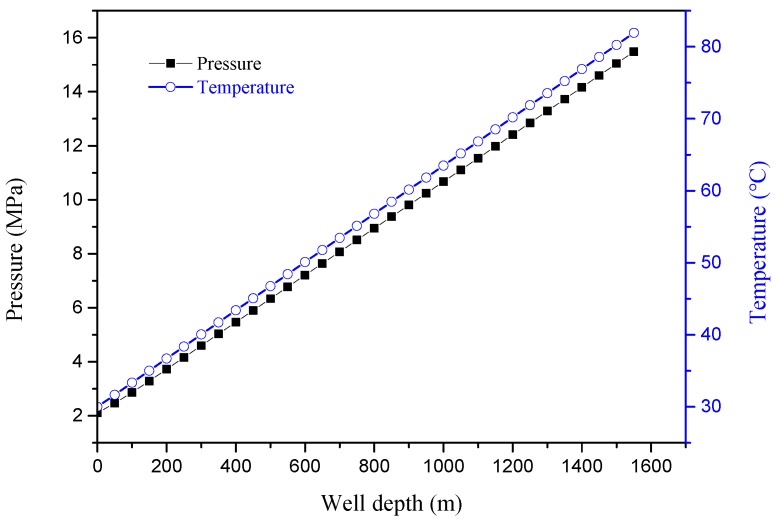
Distribution of temperature and pressure as a function of well depth.

**Figure 4 materials-11-02374-f004:**
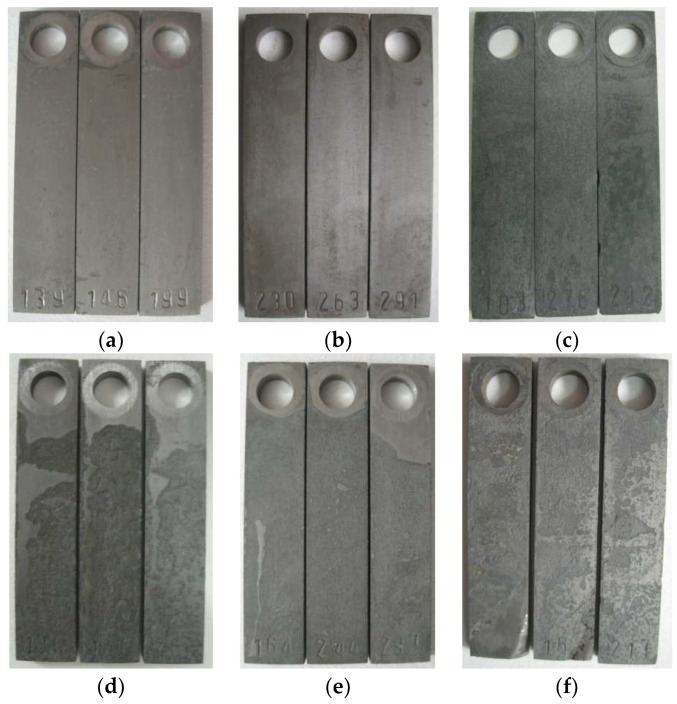
Appearance of samples corroded at the well depths of (**a**) 0, (**b**) 240, (**c**) 580, (**d**) 800, (**e**) 1050, and (**f**) 1500 m after the removal of corrosion scales.

**Figure 5 materials-11-02374-f005:**
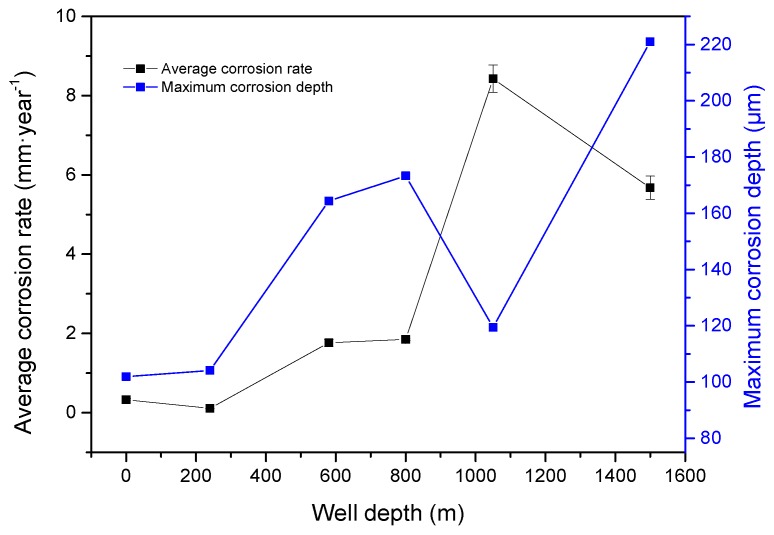
Average corrosion rate and maximum corrosion depth at different well depths.

**Figure 6 materials-11-02374-f006:**
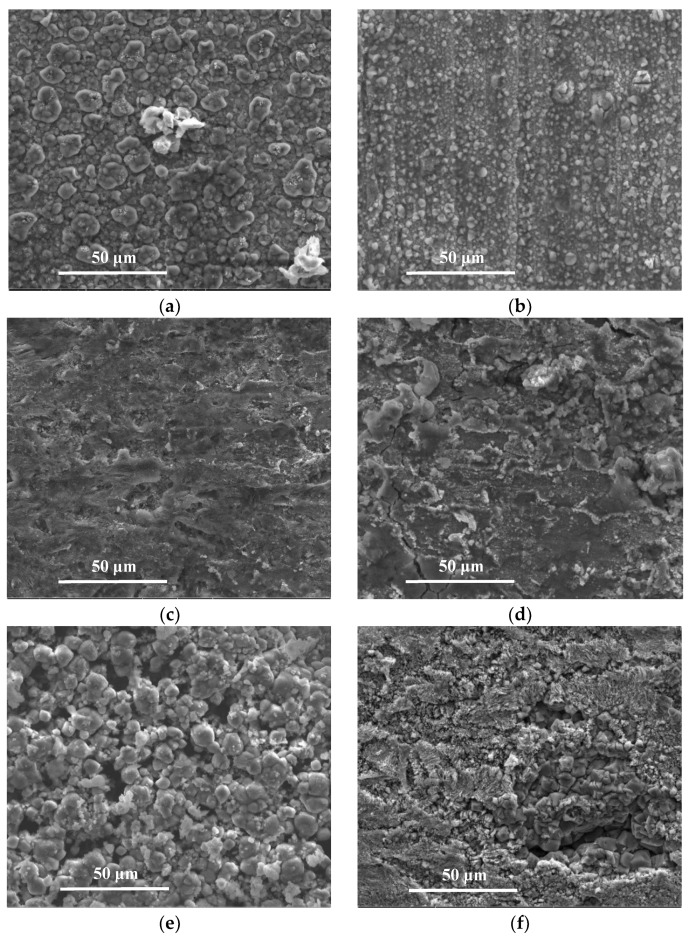
SEM images (2000×) of the surface morphologies of J55 steel after corrosion at (**a**) 0, (**b**) 240, (**c**) 580, (**d**) 800, (**e**) 1050, and (**f**) 1500 m.

**Figure 7 materials-11-02374-f007:**
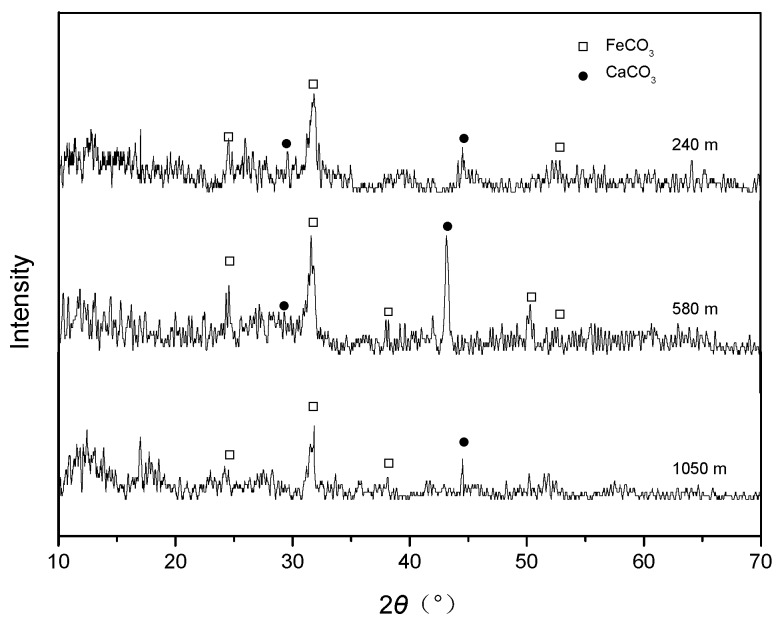
XRD spectra of the surface layers of the corroded samples.

**Figure 8 materials-11-02374-f008:**
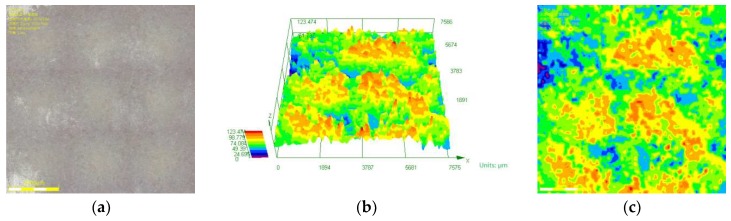
Corrosion depth analysis of the sample corroded in 30% crude oil/brine at the well depth of 800 m: (**a**) corrosion morphology, (**b**) 3D diagram of corrosion depth distribution, and (**c**) contour diagram of corrosion depth distribution.

**Figure 9 materials-11-02374-f009:**
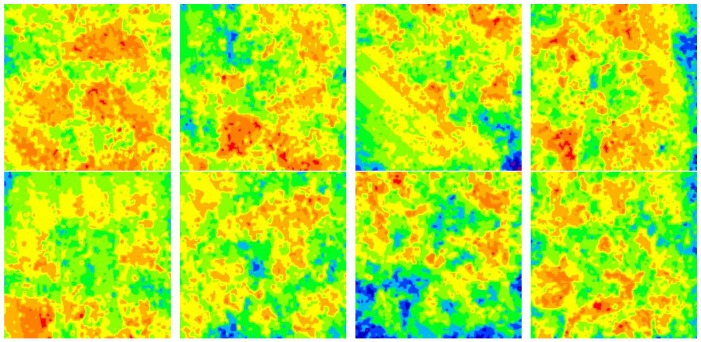
Surface height distribution of the sample corroded in 30% crude oil/brine at the well depth of 800 m after the removal of corrosion scales (100×).

**Figure 10 materials-11-02374-f010:**
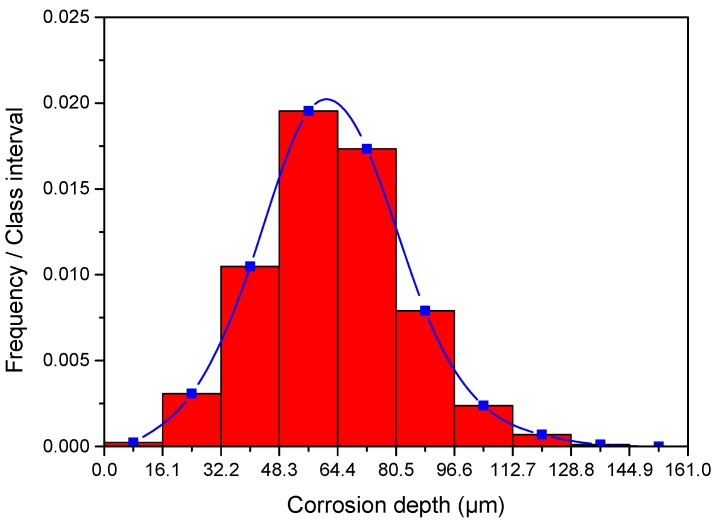
Frequency density distribution of corrosion depth on the surface of the sample corroded at 800 m after the removal of corrosion scales.

**Figure 11 materials-11-02374-f011:**
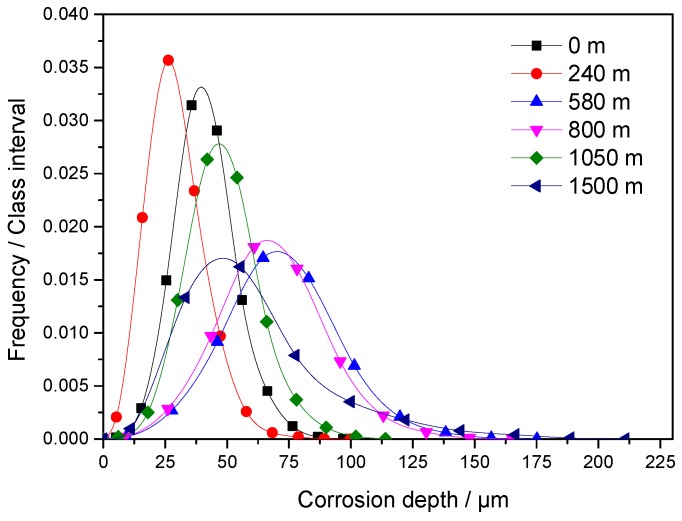
Frequency density distribution map of the corrosion depths of samples corroded at different well depths.

**Figure 12 materials-11-02374-f012:**
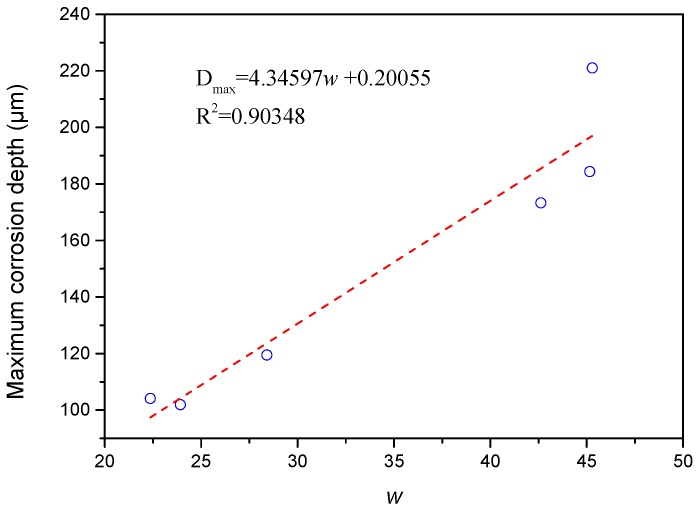
Relationship between the maximum corrosion depth and *w.*

**Table 1 materials-11-02374-t001:** Chemical composition of type J55 carbon steel (wt.%).

Elements (wt.%)								
C	Si	Mn	P	S	Cr	Ni	Cu	Fe
0.34–0.39	0.2–0.35	1.25–1.5	≤0.020	≤0.015	≤0.15	≤0.20	≤0.20	Bal.

**Table 2 materials-11-02374-t002:** Composition properties of crude oil.

Property	Units	Value
Density (20 °C)	kg·m^−3^	848.3
Kinematic viscosity (65 °C)	mm^2^·s^−1^	7.254
Acid value	mg KOH·g^−1^	0.107
Sulfur content	wt.%	0.08
Wax content	wt.%	12.86
Colloid	wt.%	2.31
asphaltene	wt.%	0.60

**Table 3 materials-11-02374-t003:** Properties of simulated brine.

Property	Units	Value
NaCl	g·L^−1^	18.5028
CaCl_2_	g·L^−1^	13.7338
MgCl_2_	g·L^−1^	0.5897
Na_2_SO_4_	g·L^−1^	0.2440
NaHCO_3_	g·L^−1^	0.0631
Salinity	g·L^−1^	33.0000

**Table 4 materials-11-02374-t004:** EDS results for the surfaces of J55 steel after corrosion (%).

	Well Depth/m	0	240	580	800	1050	1500
Element							
C K		25.04	35.63	45.36	17.94	19.14	9.92
O K		31.97	22.40	17.52	34.22	42.21	39.24
Cl K		0.26	/	/	/	0.57	1.38
Ca K		1.10	/	/	4.09	10.51	4.70
Mn K		0.69	/	0.23	/	0.54	1.39
Fe K		38.89	41.97	36.89	36.47	26.78	42.08
Cu K		2.04	/	/	6.12	/	1.29
S K		/	/	/	1.16	0.26	/
total content		100	100	100	100	100	100

**Table 5 materials-11-02374-t005:** Parameters for the Gaussian model fitting of corrosion depth distribution.

Well Depth/m	0	240	580	800	1050	1500
*x_c_*/μm	40.250	27.680	71.189	67.189	47.268	52.028
*w*	23.923	22.363	45.154	42.617	28.408	45.292

**Table 6 materials-11-02374-t006:** Results for variance analysis through Gaussian model fitting.

Statistics	DF	Sum of Squares	Mean Square	F Value	Prob > F
Global	Regression	12	8.11 × 10^−3^	6.76 × 10^−4^	719.55	0
Residual	54	5.07 × 10^−5^	9.39 × 10^−7^		
Uncorrected Total	66	8.16 × 10^−3^			
Corrected Total	60	5.11 × 10^−3^

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
