# Peer review of "Pitting Corrosion and Microstructure of J55 Carbon Steel Exposed to CO2/Crude Oil/Brine Solution under 2–15 MPa at 30–80 °C"

_materials, 2018, doi:10.3390/ma11122374_

Round 1
Reviewer 1 Report
The research manuscript contains a study aimed on corrosion properties of carbon steel J55 immersed in CO2/crude oil/brine mixture. Conditions corresponding to different wells depths were simulated and evaluated using experimental testing. Interesting statistical dependencies regarding the development of corrosion products at different well depths were observed in this study.
The submitted manuscript is interesting for readers; only minor corrections are recommended:
Line 12: correct „A This study…”
Line 75: use the same style to describe reservoirs in the manuscript, for example “Chjang-8 Oil Resevoir”
Line 79 (Table 2): “kg” instead of “Kg”
Line 83: correct “A1 L the mixture …”
The reviewer recommends the paper for publication.
Author Response
Thanks for your suggestion and advice on our paper. We have revised the manuscript according to your detailed suggestions. Words in red are the changes we have made in the text. We have revised the problems and discussed your argument as follows.
Point 1: Line 12: correct „A This study…”
Response 1: It has been changed, please see line 11.
Point 2: Line 75: use the same style to describe reservoirs in the manuscript, for example “Chjang-8 Oil Resevoir”
Response 2: It has been changed, please see line 82 and line 84.
Point 3: Line 79 (Table 2): “kg” instead of “Kg”
Response 3: It has been changed, please see line 86.
Point 4: Line 83: correct “A1 L the mixture …”
Response 4: It has been
changed, please see line 90.

Reviewer 2 Report
Dear editor, it took me more time to finish this review since I came upon a MS published on you journal by the same authors two months ago, and I wanted to compare it with the submitted one. I should also point out that the published MS was not even referenced in the submitted MS
The manuscript is entitled:
Effect of CO2 Partial Pressure on the Corrosion Behavior of J55 Carbon Steel in 30% Crude Oil/Brine Mixture
In my opinion, this is a classic salami slicing. There are differences of course e.g. the submitted one describes the corrosion of steel at different temperatures and pressure, while the published one, at one temperature and different pressure. However, the topic, solution, sample, techniques, conclusion are the same. They added the Gaussian distribution to calculate the corrosion depth of the sample. However, they could have submitted everything in one bigger MS, considering also that the experimental part is the same.
Having said that, I do not have any particular questions. The MS seems well written and presented. I only did some corrections and suggestion reported in the PDF file (e.g. misspelling or too long sentence).

Author Response
Thanks for your suggestion and advice on our paper. We have read the manuscript thoroughly and revised the manuscript according to your detailed suggestions. Words in red are the changes we have made in the text.

Reviewer 3 Report
Dear Editor,
Materials
Manuscript ID: Materials-391281
Title: Pitting corrosion and microstructure of J55 carbon steel exposed to CO2/crude oil/brine solution under 2-15MPa at 30-80 °C.
Bai et al. are studying the corrosion of carbon steel (J55) in CO2 saturated brine/oil medium under simulated conditions with defined stressed parameters (2-15 MPa/30-80 °C); magnitudes of corrosion rate for each substrate are determined by means of weight-loss measurements techniques. Surface analysis of the substrate after corrosion has been investigated using surface analytical techniques (SEM/EDX, XRD and optical microscopy). Evidence of Ca and Fe carbonate scales have been identified by XRD on corroded samples while also demonstrating surface profiling using optical microscopy. The manuscript has been read through thoroughly and some corrections must be made as indicated below to improve upon the quality of the manuscript. Authors need to read through the manuscript thoroughly and correct few grammatical errors, working tenses, grammatical errors, and spellings therein.
Reviewer
Author Response
Thanks for your suggestion and advice on our paper. We read the manuscript thoroughly and corrected few grammatical errors, working tenses, grammatical errors, and spellings. Words in red are the changes we have made in the text.
